# Predictable motion is progressively extrapolated across temporally distinct processing stages in the human visual cortex

William Turner[1,2,3]*, Charlie Sexton[3], Philippa A. Johnson[4], Ella M. Wilson[3], Hinze Hogendoorn[2,3]

1 Department of Psychology, Stanford University, Stanford, California, United States of America, 2 School of Psychology & Counselling, Queensland University of Technology, Brisbane, Australia, 3 Melbourne School of Psychological Sciences, The University of Melbourne, Melbourne, Australia, 4 Cognitive Psychology Unit, Institute of Psychology & Leiden Institute for Brain and Cognition, Leiden University, Leiden, Netherlands

* williamfrancisturner@gmail.com

## Abstract

Neural processing of sensory information takes time. Consequently, to estimate the current state of the world, the brain must rely on predictive processes—for example, extrapolating the motion of a ball to determine its probable present position. Some evidence implicates early (pre-cortical) processing in extrapolation, but it remains unclear whether extrapolation continues during later-stage (cortical) processing, where further delays accumulate. Moreover, the majority of such evidence relies on invasive neurophysiological techniques in animals, with accurate characterization of extrapolation effects in the human brain currently lacking. Here, we address these issues by demonstrating how precise probabilistic maps can be constructed from human EEG recordings. Participants ($N = 18$, two sessions) viewed a stimulus moving along a circular trajectory while electroencephalography (EEG) was recorded. Using linear discriminant analysis (LDA) classification, we extracted maps of stimulus location over time and found evidence of a forwards temporal shift occurring across temporally distinct processing stages. This accelerated emergence of position representations indicates extrapolation occurring at multiple stages of processing, with representations progressively shifted closer to real-time. We further show evidence of representational overshoot during early-stage processing following unexpected changes to an object's trajectory, and demonstrate that the observed dynamics can emerge without supervision in a simulated neural network via spike-timing-dependent plasticity.

## Introduction

Neural processing of visual information takes time. Retinal ganglion cells produce spikes at latencies of ~20–70 ms, and additional delays accumulate as these signals

**Data availability statement:** All data, as well as model simulation code can be accessed at: https://osf.io/sn4a7/. All data analysis code can be accessed at: https://github.com/bootstrapbill/neural-location-decoding. This repository has also been archived on Zenodo: https://zenodo.org/records/15283636 with DOI: 10.5281/zenodo.15283635.

**Funding:** This work was supported by Australian Research Council Grants FT200100246, DP220101166, and DP180102268 awarded to HH and Queensland University of Technology Early Career Researcher Ideas Scheme and University of Melbourne Decision Science Hub Seed Funding Grants Awarded to WT. The funders played no role in the study design, data collection and analysis, decision to publish, or preparation of the manuscript.

**Competing interests:** The authors have declared that no competing interests exist.

**Abbreviations :** FWHM, full width at half maximum; LDA, linear discriminant analysis; SEM, standard error of the mean; STDP, spike-timing-dependent plasticity; SVR, support vector regression.

pass on to downstream regions [1,2]. For time-sensitive interactions with dynamic environments, delays are problematic. For instance, imagine a hunter trying to take down bolting prey, or a tennis player trying to return a 200 km/h serve, while only having access to outdated visual information. For the tennis player, a delay of even just 50 ms will cause their ball-position estimates to be off by 2.8 m. The fact that such behaviors are still possible raises an important question: if visual processing is delayed, how do we accurately localize moving objects in real time?

Neurophysiological recordings in non-human animals have revealed clear evidence of predictive motion extrapolation occurring during the earliest stages of visual processing (see [2] for a recent review). For example, retinal ganglion cells in salamanders, rabbits, mice, and monkeys have been found to 'anticipate' the arrival of moving stimuli such that the peak of population-level activity approximately aligns with the leading edge of the stimulus despite phototransduction delays [3–7]. Similar anticipatory effects have also been observed in early cortical regions of cats [8,9] and monkeys [10–13].

Within a hierarchical predictive coding framework [14], motion extrapolation can help to minimize prediction error and thus the metabolic cost of sensory processing [15]. However, this relies upon extrapolation occurring at all stages of processing, not just the earliest, to avoid the re-emergence of delays and misalignment of sensory representations across hierarchical layers. Yet within the existing literature, extrapolatory effects have been observed predominantly in early visual regions (i.e., retina, lateral geniculate nucleus, and V1, although putative extrapolation effects have also been observed in later areas, such as middle temporal (MT)/middle superior temporal (MST) and motor regions [2,16,17]), and it is unclear to what degree neurons downstream from the retina simply inherit their extrapolated activity profiles, without actively driving further extrapolation. For example, while lesions to MT have been shown to diminish accuracy in motion prediction tasks [18], it is unclear whether this is caused by a disruption of predictive mechanisms occurring within MT or to interrupted signaling of predictive information generated in earlier regions. As such, an important open question is whether neural motion extrapolation is a multi-stage phenomenon that occurs during later-stage cortical processing.

To address this question, one path forward lies in using more global measures of neural activity to concurrently probe distinct stages of visual processing. For example, neural decoders may be trained at different time points following stimulus onset to identify and index neural activity patterns associated with distinct stages of processing (e.g., see [19,20]). To this end, anticipatory effects have recently been observed in human M/EEG [19,21–23] and fMRI recordings [24–29], with some recent evidence from our own lab suggesting that extrapolation may only occur during very early (i.e., pre-cortical) processing [19]. However, all of these past studies have been limited in their ability to examine stimulus representations with fine-grained spatial resolution, so have been unable to clearly resolve the underlying predictive dynamics.

Here, we develop a method for extracting high-resolution maps of a visual stimulus' position over time from EEG recordings. This allows us to precisely reconstruct

the trajectories of moving stimuli, revealing evidence of overshoots when stimuli disappear or reverse direction (consistent with [22]). To determine whether widespread extrapolation occurs across multiple stages of visual processing, we train machine learning classifiers on the evolving cascade of neural responses which follow the onset of static stimuli. We find evidence of the same activity patterns occurring in response to smoothly moving stimuli, but with a cumulative, compensatory shift in their timing. Specifically, activity patterns associated with temporally distinct stages of processing, which are activated sequentially under unpredictable conditions, emerge earlier than expected when viewing smoothly moving (i.e., predictable) stimuli. This accelerated emergence of position representations leads to increased temporal-alignment of representations across processing stages, reducing the gap between the encoded and actual position of the stimulus. Finally, we provide a simple, biologically plausible model that captures these dynamics, by demonstrating that they emerge spontaneously, without supervision, at all levels of a hierarchical neural network via spike-timing-dependent plasticity (STDP).

## Results

Participants ($N = 18$) each completed two experimental sessions. In each session, they viewed 2000 localizer stimuli (4,000 total), consisting of a white wedge-shaped stimulus randomly flashed in 40 equally spaced positions around an invisible circle (Fig 1A). Participants also viewed 960 smooth motion sequences in each session (1,920 total), in which the same stimulus moved along a circular trajectory for 1.5–3.5 s, before either disappearing or reversing its direction at an unpredictable location (Fig 1B). Following reversals (50% of trials), the stimulus continued moving for between 0.5 and 1 s before disappearing.

### Decoding stimulus-position information from EEG recordings

To characterize how stimulus-position information is encoded in neural activity, we first examined whether it was possible to predict the position of the static localizer stimuli from participants' neural response patterns. At each time point, we fit multivariate linear models to predict the position of a given stimulus. Because these positions are angular (i.e., circularly distributed), we trained models to predict the sine and cosine of the stimulus' angular position from the voltage at all electrodes. We scored the performance of these models by calculating the inverse absolute angular error ('decoding score') between the predicted and actual position of a stimulus (following [30], see Method).

Cross-validation revealed clear evidence of stimulus-position information in participants' neural activity emerging on average ~75 ms after stimulus onset and remaining sustained for ~500 ms ($p < .01$ cluster corrected, Fig 1C). A searchlight analysis (Fig 1C inset) indicated that a stimulus' position was best predicted from neural activity recorded over the occipital cortex. Examining average decoding performance (75–250 ms) at each localizer position, revealed that all positions could be accurately decoded, with a slight qualitative advantage for stimuli in the lower visual field (Fig 1D).

A temporal generalization analysis [31] revealed activity dynamics that were predominantly transient and evolving, with a strong diagonal response in the temporal generalization matrix and only brief, transient periods of peri-diagonal generalization (Fig 1E, $p < .01$ cluster corrected). Re-running the analysis on frequency-specific power estimates (i.e., the relative pattern of oscillatory power across electrodes at a given frequency), revealed that position information was predominantly encoded in the alpha/low-beta range (~10–20 Hz, see Fig 1F; consistent with [21]). Taken together, these dynamics are consistent with the delayed propagation of position-specific activity patterns through a hierarchical network of brain regions following stimulus onset. The predominance of a diagonal pattern within the temporal generalization matrix suggests that stimuli trigger evolving sequences of neural activity, reflecting distinct stages of sensory processing [20,31–33]. Re-running the temporal generalization analysis with a focus on sequentially predicting the location of successive stimuli (Fig 1G) revealed that information about multiple stimuli is encoded across distinct stages of processing at any given time point (consistent with [20,32]). The fact that stimulus-position information was spectrally localized in the alpha/low-beta range is in line with the recent suggestion that such rhythms may be an oscillatory 'fingerprint' of information processing within hierarchical predictive networks under neural delays [21,34].

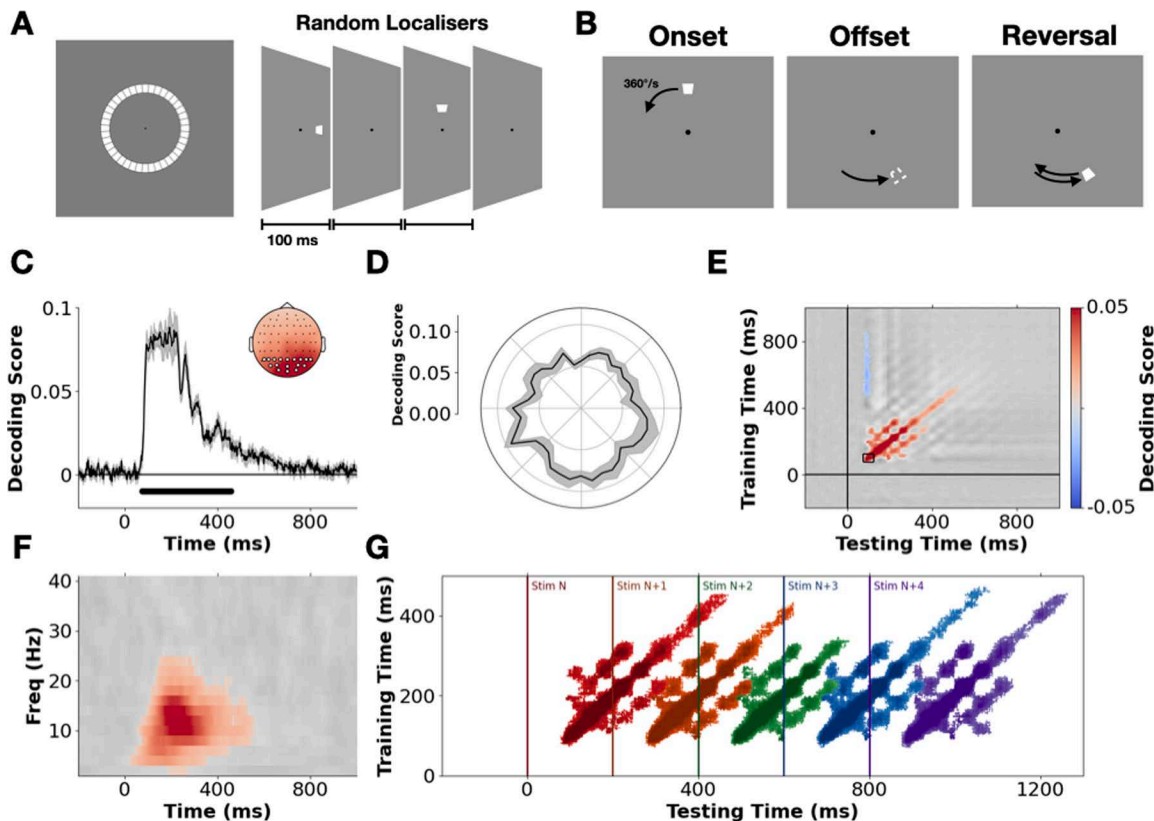

**Fig 1. Stimulus design and characterization of location-specific neural activity patterns.** (A) Localizer stimuli were randomly presented in 40 equally spaced positions tiling an invisible circle around fixation. (B) In the smooth motion sequences, participants viewed the stimulus moving along a circular trajectory at 360°/s for 1.5–3.5 s before disappearing. In 50% of trials, stimuli reversed their direction mid-way through the sequence. Following a reversal, the stimulus continued moving for 0.5–1 s. (C) Model decoding score (fractional inverse absolute angular error, 0 = guessing, 1 = perfect prediction, see Method) over time. The solid black line represents the mean group-level decoding score, with the shaded regions representing the standard error of the mean (SEM). Predictions are based on brain activity recorded at varying time points relative to localizer onset (x-axis), using data from all EEG electrodes. Black dots indicate timepoints where prediction accuracy significantly differs from chance ($p<.01$ cluster corrected, cluster forming threshold = .01, $2^{12}$ permutations). Inset shows the results of a searchlight analysis over electrodes plus their immediate neighbors. Highlighted electrodes show the posterior/occipital sites used in the subsequent LDA-based analyses. (D) Average group-level decoding performance (75–250 ms) at each localizer position, shaded regions show SEM. (E) Results of a temporal generalization analysis in which the group-level performance of time points-specific decoders is assessed across all testing time points. The full generalization matrix is plotted in grayscale with cluster-corrected timepoints overlaid in color ($p<.01$). The region marked with a black box indicates the training time-period (75–125 ms) for subsequent LDA-based mapping. (F) Results of a frequency-specific decoding analysis in which normalized power estimates were used as the input features to the model. The full results are plotted in grayscale with cluster-corrected timepoints overlaid in color. (G) Temporally generalized decoding of successive localizer stimuli. Colored regions show above chance generalization for each stimulus, respectively ($p<.01$ cluster corrected).

## Mapping the position of moving stimuli

Having confirmed that the localizers evoked position-specific activity patterns across successive hierarchical levels, we next examined whether we could leverage these patterns to continuously track the position of the stimuli during smooth motion. First, we trained multiclass linear discriminant analysis (LDA) classifiers on participants' neural responses to the localizers, treating each position as a distinct class. Then, from participant's neural responses to the smoothly moving stimuli, we extracted predicted posterior class probabilities via the pre-trained LDA models. In other words, we extracted a prediction as to the probability that the stimulus was in each localizer position at a given time point. Averaging the probabilities extracted from models trained 75–125 ms after localizer onset (see boxed region in Fig 1E), we were left with a matrix of position probabilities over time (i.e., a probabilistic spatio-temporal map).

Fig 2 shows three such maps time-locked to either stimulus onset, stimulus offset, or the reversal point in the motion sequences. Examining these, it is clear that the position of the moving stimuli can be tracked from participants' neural response patterns. From ~75 ms after stimulus onset position tuning emerges, as evidenced by the high probability region which forms around the initial position of the moving stimulus. The high probability region then shifts, drawing out a 'hockey stick' shaped pattern (reminiscent of observations in existing theoretical models, see [35,36]) and tracks the position of the moving stimulus. After motion reversal (Fig 2C), the high probability region reverses direction and continues to track the new trajectory of the stimulus, although the temporal resolution of the maps is insufficient to evaluate whether an abrupt jump or gradual shift in tracking occurs (see [7,36] for relevant considerations). The fact that we can decode a spatial probability distribution which tracks the motion stimulus is, in itself, non-trivial as smoothly moving stimuli do not evoke the well-defined onset/offset responses that have previously been leveraged to decode the position of 'apparent motion' stimuli [21,22]. The fact that we can map the position of smoothly moving objects via a bank of pre-trained static position representations indicates that the position-specific activity patterns evoked by static and dynamic stimuli overlap, at least partially (consistent with [19]).

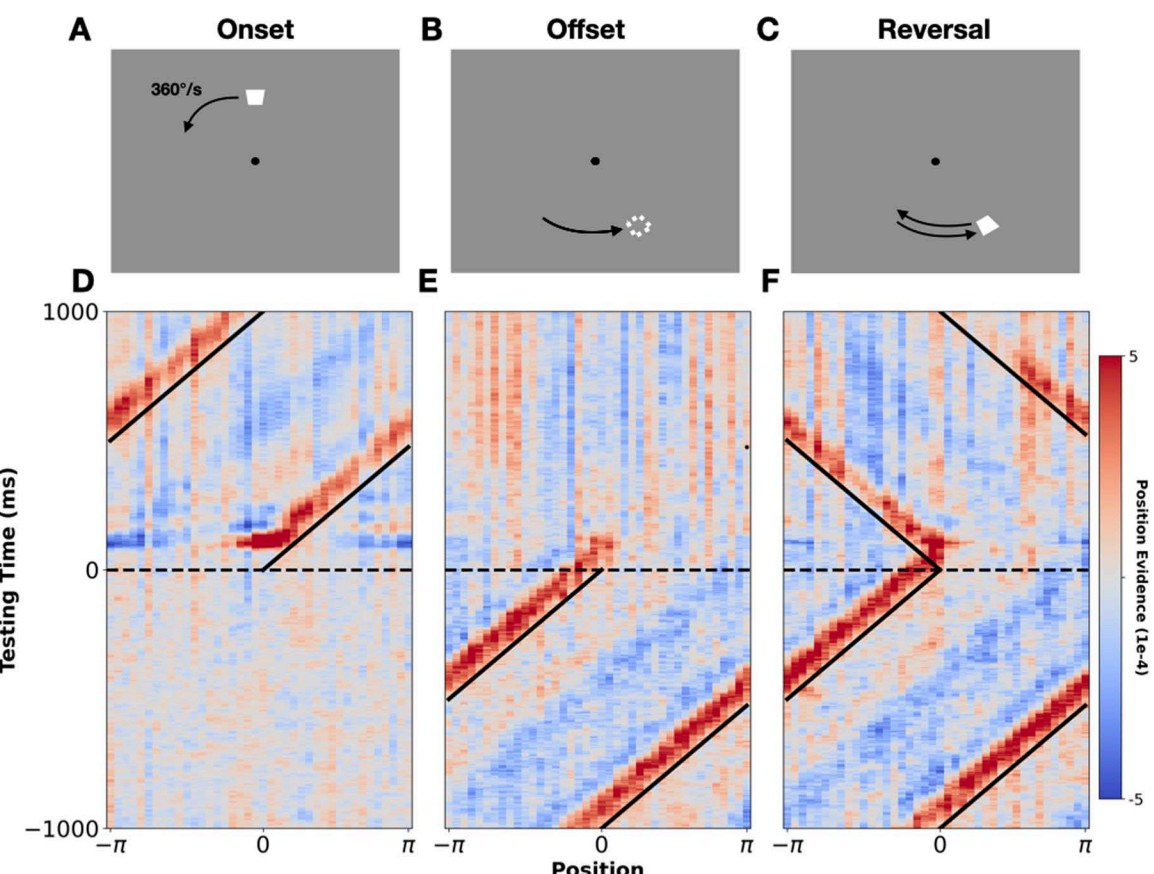

**Fig 2. Mapping the position of moving stimuli.** Panels (A–C) show the three events of interest: stimulus onset, stimulus offset, and stimulus reversal. Panels (D–F) show group-level probabilistic spatio-temporal maps centered around these three events. Diagonal black lines mark the true position of the stimulus. Horizontal dashed lines mark the time of the event of interest (stimulus onset, offset, or reversal). Red indicates high probability regions and blue indicates low probability regions ('position evidence' gives the difference between the posterior probability and chance). Note, these maps were generated from recordings at posterior/occipital sites (see highlighted electrodes in Fig 1C inset). For statistically thresholded versions of these maps, see S1 Fig.

To qualitatively examine the shape and precision of the decoded probability distributions, we took two averaged time slices: (1) directly following motion onset (75–125 ms; Figs 2), and 3A) in the lead-up to stimulus offset (−1,000 to 0 ms; Fig 3B). Examining the first (Fig 3A), we could effectively get a snapshot of the decoded probability distribution, just after visually evoked activity first reaches visual cortex. From this, we see that participants' early neural responses already encoded a remarkably precise probability distribution over space (full width at half maximum [FWHM] of above chance probabilities = 54° polar angle). Examining the second time slice (Fig 3B), we effectively get a snapshot of the decoded probability distribution, following extended exposure to motion—that is, the 'steady state' of decoded position information during motion. Notably, the distribution has narrowed (FWHM of 36° polar angle) and evolved from being more-or-less symmetric to become positively skewed in the direction of stimulus motion. The trailing edge of the distribution became

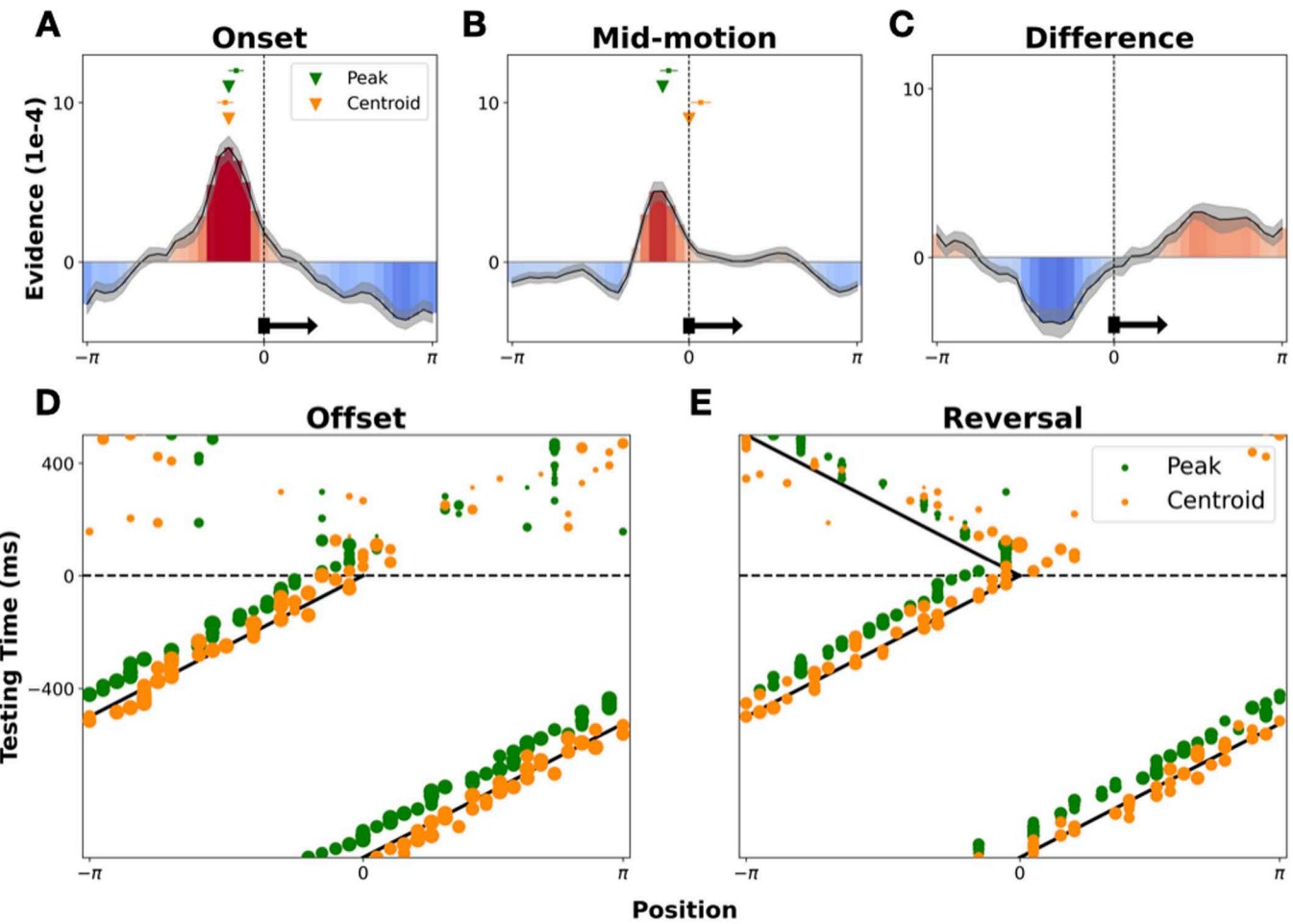

**Fig 3. Examining the shape and potential discrete read-out of decoded probability distributions.** Panels **(A and B)** show group-level averaged time slices through the maps in Fig 2, at motion onset (75−125) and mid-motion (−1,000 to 0 ms), respectively. Shaded gray sections show the standard error of the group-averaged probability distribution. Green inverted triangles show the point of peak probability, and orange triangles show the centroid (vector average) of the group-level distribution. The mean and standard error of the individual-level centroids and peaks are shown with green/orange squares and whiskers. Panel **(C)** shows the difference between the probability distributions extracted from mid-motion and immediately following stimulus onset. In panels **(A–C)** the true position, and direction, of the stimulus is marked by a vertical black dashed line and a horizontal black arrow. Panels **(D** ands **E)** show time point-specific markers of peak probability (green) and centroid (orange), overlaid on the real-time position of the stimulus around stimulus offset and reversal. Peaks and centroids are calculated every 15 ms, with the certainty of the estimate (taken as the peak height or vector average length, respectively) dictating the size of the plotted dot.

suppressed, and the leading edge became enhanced, qualitatively mirroring the changes observed in directly imaged neural activity of non-human animals (see [2]). These changes serve to shift the high probability region closer towards the real-time position of the stimulus. Since these results are generated from models trained on early neural responses to the localizers (75–125 ms), this provides evidence of extrapolation occurring during early-stage human visual processing.

Considering Fig 3A and 3B, an important auxiliary question arises: how might a point-estimate of the stimulus' real-time location be read out from these probability distributions? One option is to take the point of maximum probability (green triangles in Fig 3A and 3B show the peak of the group-level distribution, green squares and whiskers show the mean and standard error of the distribution of individual-level peaks). However, even after considerable exposure to predictable motion (panel B), this estimate continues to lag the real-time position of the stimulus (marked with the vertical black line). One-sample $t$ tests confirmed that the individual-level peaks significantly lagged the stimulus, both at the start of motion ($t = -3.49$, $p = .003$) and mid-motion ($t = -2.20$, $p = .042$). An alternative option is to take the centroid (i.e., the vector average), to better leverage information contained within the entire distribution. Interestingly, this yields an estimate (orange triangles) which initially aligns with the distribution peak (panel A, a paired sample $t$ test found no evidence of a significant difference between the peak and centroid distributions, $t = 1.44$, $p = .17$, with a one sample $t$ test indicating the individual-level centroids significantly lagged the stimulus, $t = -4.65$, $p < .001$). After ongoing exposure to predictable motion, this shifts to align with the real-time position of the stimulus (panel B, one-sample $t$ test found no evidence of a significant difference between centroids and stimulus position, $t = 1.13$, $p = 0.27$, paired sample $t$ test indicates significant difference between peaks and centroids, $t = -3.70$, $p = 0.002$). Plotting both peaks and centroids across time (Fig 3C and 3D), we can see that the peak estimate consistently lags the real-time position of the stimulus. In contrast, the centroid estimate approximately tracks the real-time stimulus position, yet overshoots when the stimulus disappears or reverses. This suggests that early visual processing may serve to encode a probability distribution over space in a manner which allows for different point estimates to be read out, depending on whether accurate instantaneous position readout is required. At any given time point, if a real-time estimate of an object's position is required, the centroid may be taken. However, if speed is not paramount (i.e., no action will be taken) the peak can instead be taken, as, after a brief delay, this will give a more reliable estimate of where the stimulus was (i.e., it will not suffer from overshoot when the stimulus changes direction). For consideration of how this speculative proposal may be quantitatively tested, see the Discussion.

## Accelerated emergence of representations across distinct processing stages

Turning to the primary question of interest, we examined how stimulus-position information was encoded across distinct stages of visual processing. Specifically, we extracted probability distributions over possible positions from decoders pre-trained on data recorded at different time points following localizer onset (Fig 4). In this way, training time effectively becomes a proxy for hierarchical level, with time point-specific decoding performance reflecting the presence or absence of position information at specific levels of representation. Then, instead of averaging across decoders (as we did to generate Fig 2), we can simultaneously consider all extracted probability distributions at once. Importantly, the temporal generalization matrix (Fig 1D) revealed a predominantly diagonal pattern, validating that decoders trained on different timepoints learn different activity patterns, thus indexing distinct stages of processing (stable activity would result in a constant, square pattern of generalization, which we do not observe, see [31]). Using training time as a proxy for processing stage, we can therefore examine how position-specific information is encoded across different stages of processing. (Note, this does not mean we can infer anything about the cortical locus of such processing, only that we are decoding from distinct activity patterns.)

To investigate the cascade of neural responses evoked by moving stimuli, after training the LDA models on localizer-evoked activity, we took 1 s epochs of EEG data in the lead up to stimulus offset or reversal (i.e., during sustained periods of smooth motion), and extracted posterior probabilities over the localizer positions from the pre-trained, time point-specific models. We then realigned the probabilities to the true position of the moving object. This allowed us to

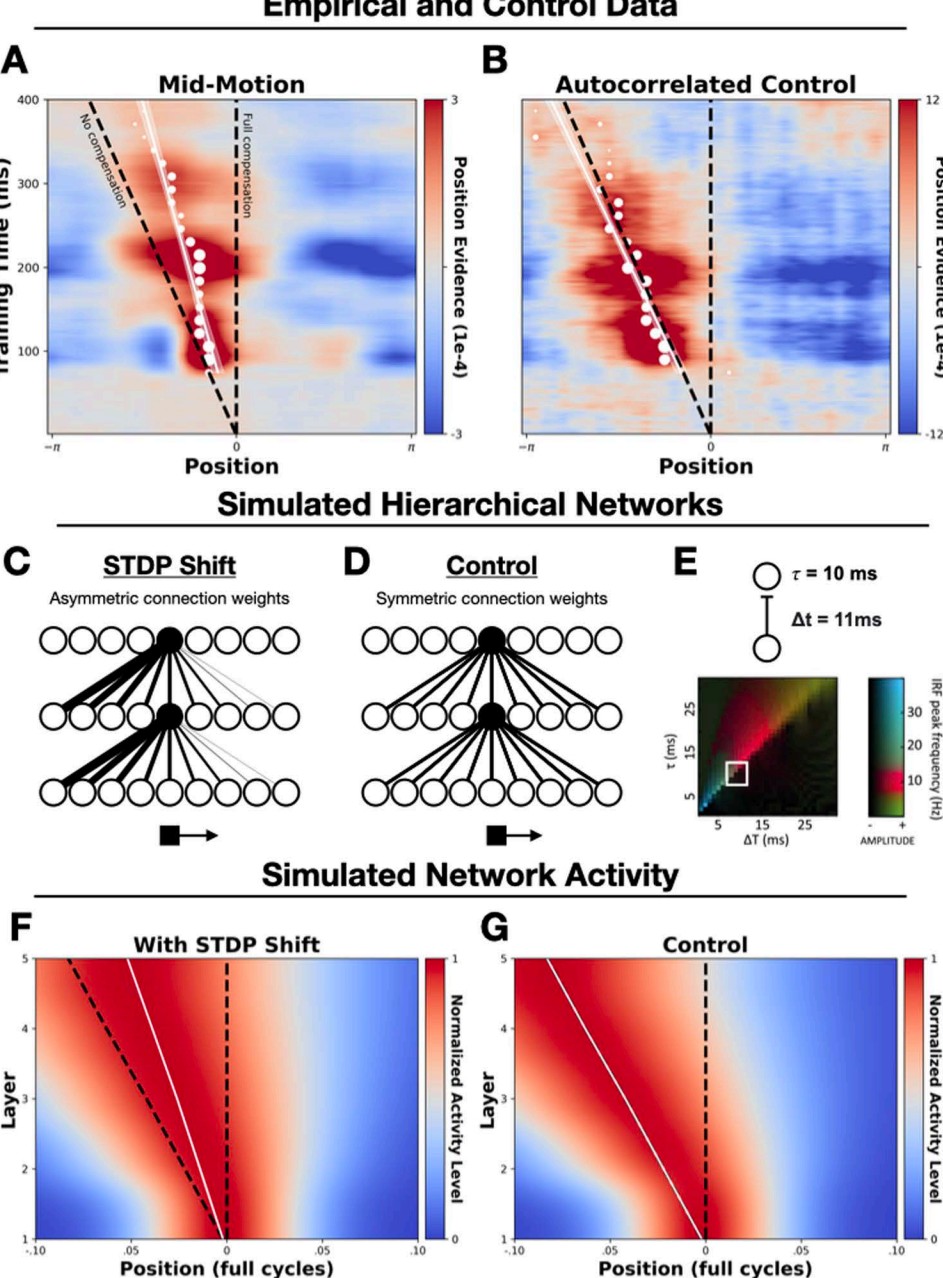

**Fig 4. Decoding position information across distinct processing stages.** Panels (A and **B)** show group-level probability distributions over spatial positions (x-axis) extracted from time point specific decoders (y-axis). White dots show the points of peak probability (in 15 ms steps), with their size being proportional to the size of the peak estimate, with overlaid regression lines and standard errors. In all panels the vertical black dashed line marks the real-time position of the stimulus ('Full Compensation'), and the diagonal dashed line marks the delayed position of the stimulus ('No Compensation'). Panels (C and **D)** show schematic depictions of the simulated STDP and control networks. In the STDP network, the receptive fields of neurons (example neurons are highlighted in black) shift in the direction opposite to motion after STDP-driven learning, allowing neurons to effectively 'anticipate' the arrival of a moving stimulus. In the control model, no shift occurs, and receptive fields are symmetrical. Panel **(E)** shows the between-layer signaling delay (Δt) and synaptic time constant (τ) used in the simulations. Also shown is Fig 1D from [34] which demonstrates that hierarchical predictive networks with delays in this approximate range (boxed white region) have oscillatory impulse response functions in the alpha/low-beta range (the range in which we found position-specific information could be best decoded). Panels **(F and G)** show simulated activity from the STDP and control networks, smoothed for visualization purposes. Plotting conventions are the same as in panels **(A and B)**. See S3 Fig for an examination of the formation of these effects following motion onset.

generate an average snapshot of how location information is encoded, at any given time, across multiple stages of visual processing during ongoing exposure to motion. If moving stimuli evoke the same cascade of neural responses as static stimuli, without any delay compensation, then the decoded position of the stimulus would sit along the diagonal line in Fig 4A and 4B ('No Compensation' line). That is, position representations encoded at later stages of processing will progressively lag those encoded at earlier stages, due to a compounding of delays as information travels along the processing hierarchy. If, instead, perfect delay compensation is achieved, then the decoded position of the stimulus would sit on the vertical line in Fig 4A and 4B ('Full Compensation' line), with the real-time position of the stimulus encoded across all processing stages.

Examining Fig 4A, we see that the bulk off the high probability region (in red) is shifted away from the No-Compensation line towards the Full-Compensation line. This indicates a shift in the timing of evoked responses, with representations emerging earlier than would be expected without delay compensation. To generate point estimates of the encoded position of the stimulus across processing stages, we overlay the time point-specific peak probabilities (in white). We chose the peak, as opposed to the centroid, as the most conservative discrete estimator, since the centroid already showed extrapolative properties (see Fig 3). However, the results are unaffected when substituting the centroid (see S2 Fig).

Examining the peak position estimates in Fig 4A and 4B, we see a sustained forward shift away from the No-Compensation line, with the distance between the points and No-Compensation line increasing over time. This is indicative of sustained extrapolation occurring, leading the encoded position of the stimulus to increasingly deviate from what would be expected as a consequence of neural delays (S3 Fig shows the formation of this predictive shift immediately following motion onset).

One reasonable concern here is that such a shift may be driven by autocorrelation of position signals at successive time points. That is, the position of a smoothly moving object is predictive of itself over short time windows. Hence, the position information within participants' neural responses will be autocorrelated. Since neighboring positions are likely to generate more similar neural activity patterns, this could conceivably blur the extracted positional probabilities. While such smearing is unlikely to affect peak probability estimates, we nevertheless sought to empirically rule out the possibility that the observed shift is simply an artifact of autocorrelation.

To this end, we ran a control analysis on 'synthetic' EEG responses to simulated motion, which we constructed by averaging across successively lagged neural responses evoked by an ordered sequence of localizer stimuli. That is, we created a matrix in which each subsequent row contained the response to the neighboring localizer, temporally offset by the time taken by the stimulus to move between these positions. Averaging across these, the resulting synthetic EEG response simulates the neural response to a localizer stimulus moving at 1 cycle per second around the display, without the presence of actual coherent motion. This allowed us to create a control condition containing autocorrelated stimulus information but no predictive dynamics (since the constituent signals were evoked by individual localizers). Analyzing these control data exactly as we did for the true motion trials (Fig 4B), we can see that the high probability region is centered along the 'no-compensation' line, demonstrating that autocorrelation cannot account for the observed spatio-temporal shift.

To statistically test for a difference in the slopes of the peak-position estimates from the actual and synthetic datasets, we fit a linear regression model to the group-level data. We predicted the peak position estimate from training time point and data type (actual versus synthetic) as well as their interaction. Crucially, the interaction term was significant ($\beta = -0.018$, SD $= .002$ $t = -7.48$, $p < .001$), indicating that the slopes of the fitted lines were different for the actual and synthetic datasets. Overall, these results suggest that the encoded position of the stimulus is shifted forwards after exposure to predictable motion, with this shift growing for later emerging neural representations. This is indicative of sustained, progressive extrapolation during cortical visual processing, resulting in the accelerated emergence of position representations and a gradual accumulation of position shifts along the processing hierarchy.

## Accounting for the observed dynamics in a hierarchical network via STDP

What might drive progressive extrapolation during cortical visual processing? Below, we provide a minimal biologically plausible model that captures the observed dynamics.

Given that we observed shifts across multiple processing stages (as indexed by time point-specific decoders), we implemented a general learning mechanism that is known to exist across cortical regions: STDP. This is a form of synaptic plasticity whereby synapses are strengthened when presynaptic cells fire shortly before a postsynaptic action potential, and weakened when they fire shortly after [37,38]. Recent research has shown how this simple associative learning mechanism can drive motion extrapolation as activity passes along the visual hierarchy [39–41]. Applying STDP to feedforward connections spontaneously produces an asymmetrical connectivity pattern, whereby the receptive fields of downstream neurons shift in the opposite direction to motion (see Fig 4C). This allows these neurons to 'anticipate' the arrival of a stimulus that is about to enter their (original) receptive field, driving a forwards shift in the population-level activity distribution (see [2] for review).

We simulated a 5-layer network with transmission delays, with feedforward connectivity profiles subject to the STDP-driven receptive field shifts reported by Sexton and colleagues [41]. The network comprised of 21 subpopulations of neurons tuned to velocities between −2 and 2 cycles/s (where negative velocities indicate counter-clockwise motion). Each level of the network comprised 1,000 neurons, with 11 ms inter-layer signaling delays and a synaptic time constant of 10 ms. Crucially, hierarchical predictive networks with delays in this general range have been found to produce activity which oscillates in the alpha/low-beta range (see [34] Fig 1D, reproduced here in Fig 4E); the same range in which we found stimulus-position information could be best decoded (see Fig 1F). We simulated firing rates across the network in response to a stimulus traversing a circular trajectory at 1 cycle/s (see the Method). We also included an initial period of velocity estimation, starting at the onset of motion, in which information about the stimulus velocity is integrated. During the earliest timepoints, activity across the velocity subpopulations is widespread, but quickly becomes centered on populations with tuning at or close to the actual stimulus velocity (see Method and S3 Fig).

Fig 4F and 4G shows maps of the activity across two simulated networks: one in which STDP-driven learning has occurred (Fig 4F) and an otherwise identical control network in which STDP-driven learning has not occurred (Fig 4G). To compare the activity of these networks to the EEG results, we computed the average activity across all neural subpopulations per level and time point—in effect extracting a macroscopic 'neural image' of the stimulus at each level of the network. Re-centering these activity profiles and averaging over time (as in Fig 4A and 4B) allowed us to then compare the position of the peak population-level response, relative to the No-Compensation and the Full-Compensation lines, in the same way as for the EEG data. Following STDP-driven learning, the activity evoked by a moving stimulus is shifted forwards off the no-compensation line (Fig 4F). Conversely, in the control model (Fig 4G) no shift occurs, with activity centered on this line.

This simulation serves to demonstrate that STDP-driven learning, and the resultant asymmetries in the receptive fields of hierarchically organized neurons, is a biologically plausible and minimally sufficient mechanism that can generate the accelerated emergence and forwards shifting of stimulus-position representations which we observed. Strikingly, these latency shifts occur entirely unsupervised. Through a known organization principle (velocity tuned sub-populations; see [15]) and local synaptic learning rule (STDP [37,42]), these dynamics emerge at all levels of the processing hierarchy, with a gradual accumulation of position shifts across levels (i.e., progressive extrapolation), mirroring what we observed in the EEG recordings of the human observers. For an additional exploratory analysis of the time course/emergence of this shift, see S3 Fig.

## Discussion

We have shown that probabilistic maps of the position of a moving stimulus can be generated from EEG recordings. We first reconstructed the trajectory of moving stimuli from early visual responses and found evidence of predictive overshoots following unexpected trajectory changes. Then, after training classifiers on the evolving cascade of neural activity patterns

which follow the onset of static localizer stimuli, we found that the same activity patterns are triggered by moving stimuli (evidenced by successful cross-generalization), but with a clear shift in their timing. Specifically, we observed an accelerated emergence of object-position representations, corresponding to a forwards shift in the neurally encoded position of the moving stimulus at higher levels of processing. As a simple, biologically plausible model of this progressive shift, we demonstrated that these dynamics emerge at all levels of a simulated hierarchical neural network via STDP.

To our knowledge, this study is the first to provide evidence of progressive, cumulative motion extrapolation during later-stage (cortical) visual processing. Evidence of neural motion extrapolation in early visual regions has previously been reported in both human and non-human animals (e.g., [3,9,10,19,22]). However, an important outstanding question has been whether motion extrapolation is a widespread, multi-level phenomenon, which continues beyond the earliest stages of processing. Indeed, while pre-cortical extrapolation mechanisms have been well-characterized [2], it has remained unclear whether cortical regions simply inherit their extrapolated activity profiles from these upstream pre-cortical regions. Our observation of a progressive shift in the encoded position of smoothly moving objects, across distinct processing stages, provides a clear answer to this question, indicating that continued extrapolation of object-position information in the cortex does occur.

A recent paper from our own lab [19] also reported evidence of extrapolation in the early visual response, but did not observe any later-stage extrapolation. However, there are several key differences in the experimental and analysis approaches across these studies that may explain this difference. Firstly, whereas the stimuli used by Johnson and colleagues moved linearly across a hexagonal grid, our stimuli moved along a circular trajectory at constant eccentricity, providing greater trial numbers and an improved signal-to-noise ratio. Secondly, stimuli in the present study moved much more quickly (~48 dva/s at its inner edge) than in Johnson and colleagues (2023; ~ 10 dva/s), and so potentially activated a largely distinct population of neurons. Finally, whereas Johnson and colleagues inspected the evolution of neural codes over time for evidence of extrapolation, here we were able to extract probability maps across space (i.e., akin to the 'neural image' of the stimulus, see [7]), allowing us to directly estimate spatial extrapolation without additional curve-fitting steps. Altogether, by fixing the eccentricity of the stimulus and collecting more repetitions of the same trajectory, we significantly improved the signal-to-noise ratio of the decoding, enabling us to adopt a more direct decoding approach and generate clear position maps (with a 5-fold increase in precision relative to our own recent attempts [21]). Given that the temporal generalization analysis provided evidence of distinct processing stages unfolding over a longer time scale, we were able to consider shifts in position representations up until 400 ms post-stimulus onset (as compared to 150 ms [19]). Overall, this revealed a clear, progressive shift in the timing of positional representations across temporally distinct processing stages.

One salient feature of the current results is that, while we find evidence of an extrapolative temporal shift, this shift is not complete. That is, peak positional probabilities never align with the real-time position of the stimulus (although centroid estimates do align with real-time in the early visual response, see Figs 3 and S2). As we have discussed elsewhere [2], it is difficult to tell whether partial shifts such as these are due to incomplete extrapolation, or rather to the fact that EEG recordings necessarily tap into a mixture of signals from different cortical regions (some of which may be fully extrapolated, and some may not be extrapolated at all). In the latter case, only signals involved in time-sensitive localization may be fully extrapolated. Ultimately, distinguishing between these possibilities will likely require the use of invasive recording procedures, where activity can be isolated to precise neural populations. What the current pattern of results does tell us is that not all position-related activity is fully extrapolated. This is sensible, as many visually evoked signals will carry auxiliary position information in addition to the actual feature they encode (since they arise from a retinotopically specific location). Because extrapolation comes with a cost (i.e., extrapolated activity must be ignored/corrected when expectations are violated), a more efficient strategy may simply be to bind featural and position information retrospectively.

Central to the question of whether partial or full extrapolation is achieved is an understanding of how the position of an object is actually read out from visual regions by downstream/effector areas. Addressing this is beyond the scope of the

current study, however, the present findings raise an interesting hypothesis. We have shown that a probability distribution over possible locations can be linearly decoded from early brain activity patterns, and that the shape of this distribution changes after exposure to motion—shifting toward the real-time stimulus position. Considering the observed changes, a hypothesis which emerges is that the shape of this distribution is modulated in such a way as to allow different point estimates of a moving object's location to be read out, depending on current task demands. Specifically, if accurate real-time interaction is paramount, then an estimate of the stimulus' real-time position may be derived from the centroid of the distribution. However, when the stimulus disappears or changes trajectory, this can overshoot and be unreliable. Conversely, when real-time interaction is less important, and accurate trajectory perception is relatively more important, the peak of the distribution may be taken, yielding a more stable, but delayed, estimate of the stimulus' position (i.e., it reliably tells you the stimulus' recent position). In other words, at any given moment, different estimators may be used, either to determine the recent position of a stimulus with high reliability (peak), or the likely current position of a stimulus, with lower reliability (centroid). Here, we note related observations made by Schwartz and colleagues [7] who observed that the peak of population-level retinal ganglion cell activity overshoots following motion reversal (see their Fig 6B). In this study, we are operating at the level of decoded probability distributions, so we cannot confidently assert anything about the underlying distribution of neural activity. Nevertheless, one possibility is that transformations occur as information passes along the visual hierarchy, gradually removing the retinal overshoot, eventually aligning with observations of smooth reversal trajectories on the conscious perception level (see [36,43–45]). To further test whether such transformations are taking place and whether early cortical regions encode a positional probability distribution from which different discrete estimates can be read out according to task demands, future studies will need to vary the speed of the moving stimulus. The central test will be whether the distribution centroid shifts to align with the real-time position of the stimulus regardless of its speed, serving as an effective real-time estimate. If this occurs, then future studies would also need to determine whether/how downstream regions can extract summary measures (such as a centroid) of population-level activity in early visual regions, and ultimately how these relate to behavior (such as reaching or related targeting actions). Relatedly, future studies should also attempt to determine the specific neural mechanisms underlying the observed distribution shift. This would help to rule out any concern that this shift trivially arises from the fact that motion onset responses are more similar to localizer evoked activity than mid-motion responses, although this concern is already reduced by the fact that the shift is conditional on the direction of the stimulus, and is consistent with direct observations of neural activity in animals (see [2] for a recent review). Indeed, taking inspiration from this prior animal work, studies could examine the potential roles of laterally spreading activity [4,6], suppressive traveling waves [46], and related mechanisms, in dampening activity at the trailing edge and/or enhancing activity at the leading edge of a moving stimulus, driving a distribution shift. For instance, it may be interesting to examine the degree to which anticipatory effects are abolished during visual midline transitions, as laterally spreading activation will be disrupted, with signals instead passing through the corpus callosum (e.g., see [10]).

By simulating the activity of a hierarchically organized network of neurons, we showed how STDP-driven learning can drive the accelerated emergence of object-position representations, in an unsupervised fashion. Our main aim in presenting this result is to show how a simple cortical mechanism can, in principle, drive the sustained, progressive extrapolation we observed. Indeed, given the existence of velocity-tuned neural subpopulations, and the ubiquity of STDP-based learning throughout the cortex, the simulations provide compelling support for our empirical findings, as they suggest that something additional would be needed to *prevent* such temporal shifts from occurring. Crucially, these shifts will only occur for sequences of input to which we are frequently exposed (like smooth motion), and will also depend on the specific plasticity of the underlying sensory region.

Importantly, in presenting the network simulations, our intention is not to claim that STDP is the sole driver of neural motion extrapolation. In prior work, a variety of other extrapolation mechanisms have been well catalogued, with many operating during pre-cortical visual processing (see [2]). These can drive forwards shifts in the evoked population-level distribution of neural activity, and may well have contributed to the motion-induced shape changes we observed in the

probability distributions decoded from the early visual response (i.e., <150 ms). As such, we caution against taking the STDP network model as a complete model of neural extrapolation. Nevertheless, given its generality, we feel that STDP is an important candidate mechanism to consider—especially when accounting for widespread, progressive effects such as those we observed. Indeed, given the ubiquity of STDP, an interesting avenue of future research may involve examining whether sustained temporal shifts in the pattern of evoked neural responses can be found after ongoing exposure to 'motion' through more abstract feature space, such as color, luminance, or numerosity. Moreover, it is worth noting that our focus has been on smooth first-order motion, but future studies should examine whether the current observations hold for alternative forms of motion (e.g., second-order motion, biological motion, or optic flow) which rely on at least partially distinct processing mechanisms. Finally, a remarkable feature of the current model (and [40,41]) is that spontaneous motion extrapolation is achieved during purely feed-forward processing. However, future studies should consider further developing these models, building in recurrent and/or horizontal connections. These may act as putative mechanisms for fine-tuning the magnitude of STDP-driven extrapolation in a delay-dependent fashion (i.e., calibrating the degree of shift to a given delay). Of potential relevance here is the fact that we see oscillatory patterns of peri-diagonal rebounds in the temporal generalization matrix (Fig 1E), with a frequency-specific decoding analysis showing that position information is predominantly localized in the alpha/low-beta range (Fig 1F). This is consistent with recent work from our lab showing evidence that oscillations in this same range carry predictive position information [21], in line with theoretical work from Alamia and VanRullen [34] suggesting that such oscillations may be a spectral signature of hierarchical predictive coding. However, since the localizer stimuli in the current study were presented at a harmonic of alpha (5 Hz), it is difficult to disentangle the degree to which these effects are endogenously or exogenously driven. As such, examining the potential oscillatory underpinnings of these effects, and using these dynamics to constrain/determine the possible underlying neuro-computational architectures (see [20,47]) remains an exciting avenue for future research.

Finally, it is important to consider the possibility that the observed effects were driven by eye-movement-related confounds. For three main reasons, we are confident this is not the case. Firstly, the results of our searchlight analysis (Fig 1C inset) demonstrate that decoding is being driven by activity from occipital electrodes, over the visual cortex. This is in contrast to previous work showing that eye-movement-related confounds in neural decoding analyses typically manifest over frontal regions (see Fig 6 in [48]). Secondly, while we did not employ an eye tracker, we explicitly instructed participants not to move their eyes to the stimuli and used a fixation target which has been shown to best reduce eye movements and promote fixation [49]. Given that the stimuli were presented at a distance of 7.7–9.4 dva from fixation (from inner to outer edge), with the static stimuli presented at a rapid temporal rate (5 Hz) and the dynamic stimuli moving at 48 dva/s, it is unlikely that participants would have been able to continually and reliable fixate on the stimuli. Finally, for the main LDA-based mapping analyses, we restricted our focus to only occipital electrodes and trained the decoding models only on neural activity patterns 75–125 ms after the onset of the localizer flashes. Since saccade onsets, and the resulting eye-movement-related confounds in decoding performance, typically occur after >200 ms [48], this further reduces the likelihood of eye-movement confounds explaining our results. The fact that we trained on such early time points, and generalized models from flash-evoked activity to motion-evoked activity, also reduces concerns about insidious confounding driven by secondary effects of eye movements (e.g., changes in visual cortex activity driven by small eye movements and the resulting movement of stimuli across the retina), as it is unlikely that the static and moving stimuli will have reliably triggered common small eye movement patterns, particularly during such early time windows.

In sum, we have shown how precise probabilistic maps of the position of a moving object can be generated from EEG recordings. Using this approach, we have provided clear evidence of progressive neural motion extrapolation occurring during visual processing in the human brain. Most strikingly, we have shown that after ongoing exposure to smooth motion, there is an accelerated emergence of position-specific activity patterns across distinct processing stages, corresponding to a forwards shift in the neurally encoded position of the moving stimulus. This provides the first clear evidence of cortical neural motion extrapolation in the human visual system. Finally, we have shown how these dynamics would be

expected to emerge spontaneously, without supervision, at all levels of a hierarchical neural network via STDP—providing a mechanism for widespread neural extrapolation/delay compensation.

## Method

### Ethics statement

The experimental protocol was approved by the human research ethics committee of the University of Melbourne (Reference Number: 2021-12985-16726-4), and the study was conducted according to the principles expressed in the Declaration of Helsinki. All participants gave written informed consent prior to participating.

### Participants

Eighteen observers (15 female, 18–35 years old with a mean age of 23 years) participated in the experiment. Each observer completed two sessions across separate days. All had normal or corrected-to-normal vision, gave written informed consent at the beginning of each session, and were reimbursed AUD15 per hour.

### Stimuli

Stimuli were generated using the Psychophysics Toolbox (Brainard, 1997) in MATLAB 2016a (Mathworks). Stimuli were presented on an ASUS ROG PG258 monitor with a resolution of 1,920 × 1,080 running at 120 Hz. The stimulus was a white, truncated wedge presented on a uniform gray background (Fig 1). The inner and outer edges of the wedge were 7.7° of visual angle (dva) and 9.4 dva away from fixation. The wedge covered 9° of polar angle with 1.19 dva at the inner and 1.46 dva at the outer edge. During localizer trials, the stimulus could appear in one of 40 locations tiling an invisible circle centered on the fixation point (see Fig 1A). Localizer stimuli were presented for 100 ms, with an interstimulus interval of 100 ms (i.e., onset rate of 5 Hz). Smooth motion sequences began and ended in randomly determined localizer positions, with sequences separated by an interval of 500 ms. The smoothly moving stimulus had a velocity of 360° of polar angle per second (i.e., a 3° offset per frame).

### Task

Participants viewed the stimuli while EEG was recorded. In the localizer block, participants viewed the stimulus being randomly presented in 40 equally spaced positions around fixation (50 repetitions per position, per session). In the smooth motion block, participants reviewed the same stimulus moving for at least 1.5 s before either disappearing or reversing direction at a randomly determined localizer position. Following reversals the stimulus continued to move for 0.5–1 s. Participants viewed 960 motion sequences per session (12 repetitions per position and motion direction). Participants were given a self-paced break halfway through the localizers and five self-paced breaks during the smooth motion sequences. To maintain participants' attention, they were tasked with pressing a button whenever the stimulus changed from white to purple. This occurred 20 times during the localizer block and 50 times during the smooth motion block. Neural responses to these 'catch' stimuli were not analyzed. The order in which participants viewed the localizer and smooth motion blocks was randomized in each session.

### EEG acquisition and pre-processing

64-channel EEG data, as well as data from six EOG electrodes (placed above, below, and next to the outer canthi of each eye) and two mastoid electrodes, were acquired using a BioSemi ActiveTwo EEG system sampling at 2,048 Hz. EEG data were re-referenced offline to the average of the two mastoid electrodes and resampled to 512 Hz. Bad channels noted during data collection (mean of 1 per session, max of 3) were spherically interpolated using the MNE 'interpolate_bads' function [50].

For the support vector regression (SVR)-based decoding analyses (used to initially characterize the neural encoding of position-specific information), we extracted epochs of data (−200 to 1,000 ms) relative to localizer onset. These were baseline corrected to the mean of the 200-ms period prior to stimulus onset. For the LDA-based mapping analyses, we draw a distinction between training and testing epochs. Training epochs were extracted from −200 to 400 ms relative to localizer onset, and were baseline corrected (−200 to 0 ms prior to stimulus onset). Testing epochs were extracted from the smooth motion sequences (−1,000 to 1,000 ms) relative to the events of interest (onset, offset, reversal), and were baseline corrected to the mean of the preceding 1,000 ms period (i.e., one full cycle of motion). For the time-frequency based decoding analyses (see below), power estimates were extracted at 20 linearly spaced frequencies between 2 and 40 Hz using the tfr_morlet function in MNE, with the number of cycles, which are used to define the width of the wavelet's Gaussian window (n_cycles/2*pi*f), logarithmically increasing from 3 to 10 across frequencies.

## Decoding analyses

Principal component analysis was applied before decoding to capture 99% of the variance (transformation computed on training data and applied to testing data), to help de-noise the data [51]. Below we outline the details of the two decoding approaches we have employed, using SVR and LDA models, respectively. Prior to this, however, we will briefly outline the goals behind each of these approaches and logic behind choosing these specific models.

For our initial analyses (Fig 1), the aim was to search for and characterize sources of stimulus-position information within the neural signal, which could then be leveraged for a subsequent mapping analysis. In doing so, we sought to use an approach which would be maximally sensitive to any sources of information, while also being computationally efficient. Following King, Pescetelli, and Dehaene [30], we chose to perform circular SVR. This approach respects the circular nature of the outcome variable and allows us to efficiently derive a continuous measure of prediction accuracy (inverse angular error). In theory, we could have used LDA, however, to avoid overly harsh model evaluation we would need to adopt more computationally intensive procedures (i.e., 'out of the box' this is a discrete prediction algorithm, with prediction accuracy tied to correctly determining class membership in a 1/40 classification problem). For example, we would have to generate a probabilistic map of the localizer stimuli and then convolve a cosine function with these maps to derive an estimate of position tuning (e.g., [52]). For the time-generalized and frequency-specific analyses (Fig 1E–1G), this would rapidly become computationally costly and unwieldy (i.e., we would need to repeat this process for each pixel within those subplots). Hence, for the sake of computational efficiency, we chose to use angular SVR for these analyses.

For the mapping analyses (Figs 2–4), the benefit of switching to use LDA is that it allows us to generate a probabilistic prediction for each possible position simultaneously. In theory, we could have used SVR or logistic regression to attempt to conduct the same analysis (e.g., by counting the frequency with which positions are mis-classified, and plotting the mis-classification frequencies in a map, see [20]). However, this is sub-optimal as at each time point only a single prediction (as opposed to a prediction for each class) is generated, making it a less efficient use of the data.

**SVR-based analyses.** To initially characterize the neural encoding of stimulus-position information we trained multivariate linear models (SVR with L2 loss) to predict the position of the localizer stimuli from participants' neural activity patterns across all electrodes (following [30]). Specifically, we trained models to predict the sine and cosine of the angular position of the stimulus. A custom scoring function was used to calculate the fractional inverse absolute angular error ('decoding score') between the predicted and actual position of the stimulus:

$$Decoding\ score = \frac{2\left(\frac{\pi}{2} - \frac{1}{n}\sum_{j=1}^{n}\left|\arg\left(\frac{e^{i\hat{y}_j}}{e^{iy_j}}\right)\right|\right)}{\pi} \tag{1}$$

where $\hat{y}_j$ is the predicted angular position on trial $j$, and $y_j$ is the actual angular position of the stimulus on trial $j$. This is designed such that a score of 0 indicates chance performance and a score of 1 indicates perfect accuracy. Custom 5-fold

cross-validation was used to evaluate out-of-sample prediction accuracy, ensuring no leakage between test and training sets.

Temporal generalization analysis was conducted by examining how well models trained on neural activity patterns at one specific time point could predict the position of stimuli based on data from other time points (see [31]). A searchlight analysis across the scalp was conducted by running the decoding analysis using data from a single electrode plus its immediate neighbors (following [52]). The spectral locus of position information was examined by re-running the decoding analysis on frequency-specific normalized power estimates (i.e., the relative pattern of oscillatory power across electrodes in a given frequency band, see [21]). Finally, to examine how successive stimuli were encoded (following [20,32]), we re-ran the temporal generalization analysis after splitting the localizer data in half. From one-half, we extracted training epochs (0–500 ms) relative to the onset of each localizer. From the other half, we extracted testing epochs (−200 to 1,300 ms) relative to every fifth localizer stimulus. We then iteratively predicted the spatial location of these stimuli followed by the locations of the four subsequent stimuli. Finally, we re-ran the analysis after switching training and testing sets.

**LDA-based mapping.** To map the location of the smoothly moving stimuli, we trained multiclass LDA classifiers on individual participants' neural responses to the localizer stimuli, treating each position as a distinct class. For these analyses, we restricted our focus to the data recorded from the 17 occipital parietal electrodes (P7, P5, P3, P1, Pz, P2, P4, P6, P8, PO7, PO3, POz, PO4, PO8, O1, Oz, and O2; see Fig 1C inset). We extracted predicted posterior class probabilities from these pre-trained LDA models (via the 'predict_proba' function in scikit-learn), based on participants' neural responses to the smoothly moving stimuli. Averaging the posterior probabilities across models trained 75–125 ms after localizer onset yielded a single matrix of probabilities (i.e., a spatio-temporal probabilistic map of the stimulus' position over time). To calculate the centroid at a given time point, we took a weighted vector average:

$$\overline{a} = \frac{1}{n} \sum_{j=1}^{n} w_j \cdot \exp(i \cdot a_j)$$

(2)

where $j$ indexes position, $w_j$ is the posterior probability at position $j$, $i$ is the imaginary operator, and $a_j$ is the polar angle of position $j$ (in radians). The centroid was then taken as the argument of $\overline{a}$. In visualizing the results of the LDA-based analyses (Figs 2–4), posterior probabilities are converted to 'position evidence', a measure of the difference between a given posterior probability value and chance (i.e., the probability of correctly guessing the stimulus' position, which is 1/40 or 0.025).

## Statistical analyses

For the SVR-based decoding analyses (Fig 1), we performed cluster-corrected one-sample $t$ tests (two-tailed) against zero using a critical alpha level of .01 and a cluster forming t-threshold corresponding to a $p$-value of 0.01 (computed via the MNE function 'spatio_temporal_cluster_1samp_test', with $2^{12}$ permutations). For the statistical thresholding of the LDA-derived probabilistic maps (S1 Fig), we conducted one-sample $t$ tests (one-tailed) at thresholds of $p < .05$ and $p < .01$. To statistically test for differences between the distributions of peaks and centroids extracted from individual-level data, and between these distributions and the true position of the stimulus, we conducted paired sample and one-sample $t$ tests (two-tailed), respectively. To statistically test for a difference in the slopes of the peak-position estimates from the actual and synthetic datasets in Fig 4, we fit a linear regression model to the group-level data. We predicted the peak position estimate from training time point and data type as well as their interaction.

## Neural network modeling

To simulate representations of stimulus position at different stages of cortical processing, we used a hierarchical network similar to that described by Sexton and colleagues [41] (see [40] for a two-layer implementation). The network consists of $N_l$ layers, each comprising $N_v$ velocity-tuned subpopulations. Within each layer and velocity subset, there are $N_n$ neurons

with spatial tunings distributed across a circular interval [0, 1], each receiving feedforward activity from neurons in the layer below (excepting the first layer, which encodes the stimulus position). The feedforward weights $W$ for each neuron follow a Gaussian distribution with width $\sigma_w$ and mean $\mu_w$:

$$\mu_w = \; x_i^l + \beta STDP_{vl} \tag{3}$$

where $x_i^l$ is the spatial position of the neuron $i$ at layer $l$, $\beta$ is the STDP shift scaling parameter, and $STDP_{vl}$ is the magnitude of the receptive field shift for the velocity $v$ and layer $l$, as taken from [41] (see Estimating STDP-driven receptive field shift magnitudes). The addition of the $STDP_{vl}$ term in Eq. (3) means that the distribution of feedforward activity does not remain symmetric as it progresses through each layer, but is shifted in line with the receptive field shift magnitude. We included the scaling parameter $\beta$ in order to find STDP shift magnitudes that best fit the EEG data (see Fitting procedure). Firing rates are recorded during a simulation carried out across $N_t$ time points, during which a point stimulus traverses a circular trajectory at 1 cycle/s. The stimulus position is encoded at each timestep $t$ by the firing rate of neurons in the first (input) layer, $r_v^1(t)$, described by a Gaussian distribution centered on the stimulus position and with width $\sigma_p$. The input layer neurons have a baseline firing rate $r_b$, with stimulus magnitude equal to $Sr_b$. Unlike in Sexton and colleagues (2023) [41], no spikes are generated: only firing rates are transferred between layers, subject to a transmission delay $t_{delay}$.

Firing rates at each higher layer are based upon (delayed) input from the layer below:

$$r_v^l(t) = \; r_v^l(t - \Delta t)e^{-\frac{\Delta t}{\tau_m}} + \; \frac{\Delta t}{\tau_m}I_v^l(t) \tag{4}$$

where $\tau_m$ is the passive membrane time constant, $\Delta t$ is the length of the timestep used, and $I_v^l(t)$ is the delayed input to the velocity subpopulation at this layer:

$$I_j^l(t) = \; W_v^{l-1} \cdot \; r_v^{l-1}(t - t_{delay}) \tag{5}$$

Following the simulation, a global estimate of the represented stimulus position is generated for each time point and layer, by taking a weighted average of firing rate distributions across all velocity subpopulations. The weights for each velocity are time-dependent: at motion onset activity is widespread across all velocity subpopulations, before becoming primarily dominated by the neural activity tuned to the stimulus velocity. (Note: this assumption is for the purposes of the analyses presented in S3 Fig, and the results of the main analysis do not hinge upon it.)

Specifically, we defined a range of time points [0, $T_i$] in which information about the stimulus velocity is integrated. Starting at $t = 0$, weights across velocity subpopulations are generated according to a Gaussian distribution centered on $v = 0$, with width $\sigma_g(t = 0)$. At each subsequent timestep, the mean of the Gaussian, $\mu_g(t)$, is shifted in the direction of the true stimulus velocity by an interval such that $\mu_g(t = T_i)$ is centered on the true stimulus velocity. Likewise, $\sigma_g(t)$ is decreased incrementally across each of the integration timesteps (see Fig 5). Because information about stimulus velocity is also subject to transmission delays, the change in weights thus described is applied to each layer in a delayed manner. The weighted average of firing rates across all velocity subpopulations is calculated for each layer and time point, then normalized, to generate a final estimate of the global position representation at each layer and time point.

The precise parameter values used in the simulation are shown in Table 1, and were taken from Sexton and colleagues [41] where applicable.

**Estimating STDP-driven receptive field shift magnitudes.** The STDP shift magnitudes estimated by Sexton and colleagues [41] are reported in the range 0.1–1 cycles/s, as well as 2, 3, 4, and 5 cycles/s. We wished to include a range of velocities that was symmetric around zero, and for which the EEG experiment velocity used (1 cycles/s) was an intermediate value, to avoid any edge effects during averaging. Therefore, the values reported by Sexton and colleagues

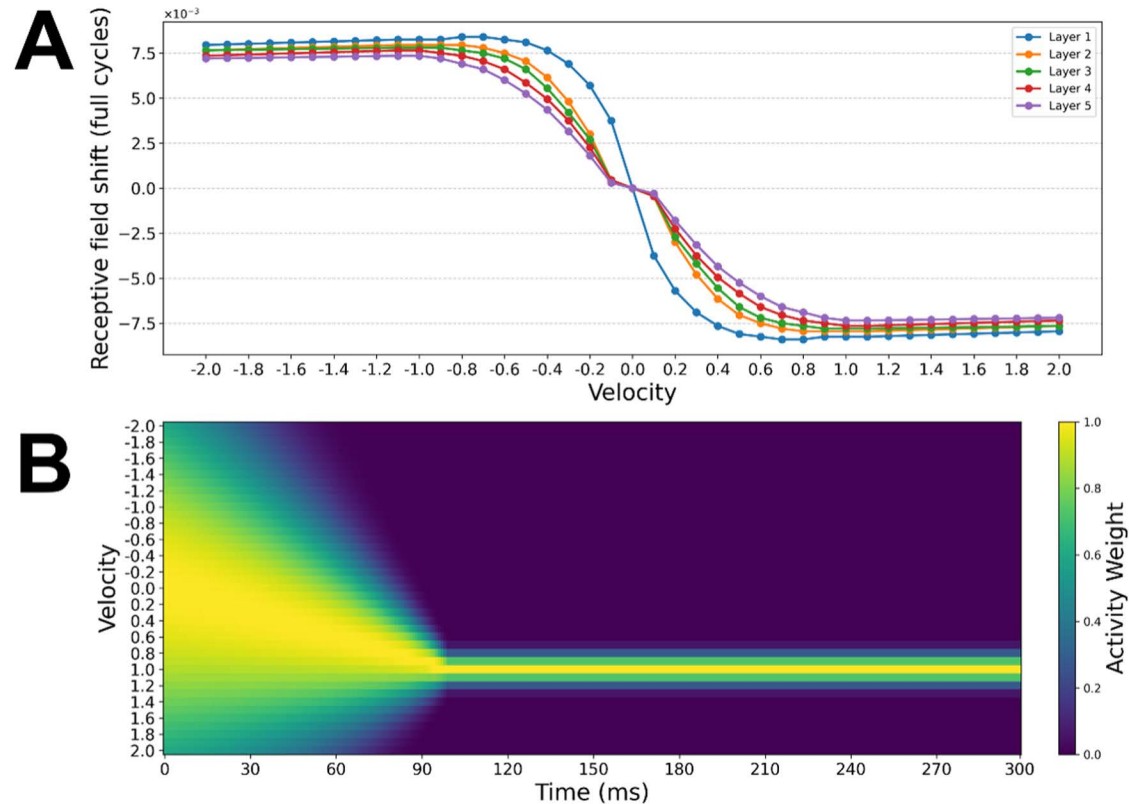

**Fig 5. Contribution of velocity-specific receptive field shifts to global position representations.** (A) Magnitude of STDP-driven receptive field shifts across the velocity range −2 to 2 cycles/s, estimated based on the shifts reported by Sexton and colleagues [41] (see Estimating STDP-driven receptive field shift magnitudes). Individual lines show STDP shift magnitudes for each layer of the network. (B) Temporal evolution of activity weights for each velocity subpopulation during the simulation. At onset, the activity is broadly tuned around 0 cycles/s, then shifts toward the true stimulus velocity (1 cycle/s) during the velocity integration period. Following the initial integration period, activity weights remain stable and centered on the true stimulus velocity.

**Table 1. Parameter values used in numerical simulation.**

| Name | Value | Description |
|---|---|---|
| $N_l$ | 5 | Number of layers |
| $N_n$ | 1,000 | Number of neurons per layer |
| $N_v$ | 21 | Number of velocities |
| $\sigma_w$ | 1/32 | Standard deviation of anatomical connections |
| $\sigma_p$ | 1/32 | Standard deviation of stimulus input |
| $r_b$ | 5 Hz | Baseline firing rate at input layer |
| $S$ | 20 | Stimulus intensity (a.u.) |
| $\Delta t$ | 1 ms | Timestep length |
| $N_t$ | 300 | Total simulation timesteps |
| $T_i$ | 100 | Velocity integration timesteps |
| $\tau m$ | 10 ms | Membrane time constant |
| $t_{delay}$ | 11 ms | Transmission delay |
| $B$ | .6 | STDP shift scaling parameter |
| $\sigma_g(t=0)$ | 2 | Standard deviation of activity weights at onset |
| $\sigma_g(t=T_i)$ | 1/8 | Standard deviation of activity weights at $T_i$ |

[41] were extended by linearly interpolating between the values given for 1 and 2 cycles/s, and then inverting all positive values to generate a mirrored set of velocities tuned to the opposite directions. This allowed us to generate an estimate of STDP magnitudes for velocities in the range from −2 to 2 cycle/s, in 0.1 cycle/s increments (see Fig 5).

**Fitting procedure.** To compare the simulated network to the position representations decoded from EEG, we performed the recentering of the global position representations such that the real-time position of the stimulus is always centered on the midpoint, with lagged representations indicated by activity at any locations counter-clockwise of this central position. The resulting map plots position relative to stimulus on the horizontal axis and network depth (layer number) on the vertical axis.

As with the EEG data, we can define two lines within this rotated map of positional representations: (1) a diagonal 'No Compensation' line, connecting the points at each layer/training time point where the stimulus would be represented in the absence of any extrapolation, given the stimulus velocity, temporal lag between layers and the compounding effect of the synaptic time constant on peak activity, and (2) a vertical 'Full Compensation' line representing the real-time stimulus position (perfect delay compensation/extrapolation). The degree of extrapolation can therefore be quantified by measuring where peak activity sits as a ratio of the difference between the No-Compensation and Full-Compensation lines (Fig 6A). This ratio measurement provides us with a simple, straightforward way of comparing the degree to which positional representations are shifted in the EEG and simulated networks.

In the simulated network, the extrapolation magnitude for a given velocity is largely determined by the transmission delay and time constant (affecting primarily the position of the No-Compensation line, relative to the Full-Compensation line), and the magnitude of the STDP shifts (affecting primarily the position of the neural representation line, relative to the No-Compensation line). We ran the simulation while varying both transmission delay ($t_{delay}$; 5–25 ms) and STDP shift magnitude (via the scaling factor $\beta$; .1 to 2 times the values reported in Sexton and colleagues [41]), to find parameter values which led to an extrapolation magnitude that best fit the ratio derived from the EEG data (.36). The extrapolation magnitude measurement was made based on activity recorded at the final time point of the simulation, and at 400 ms for

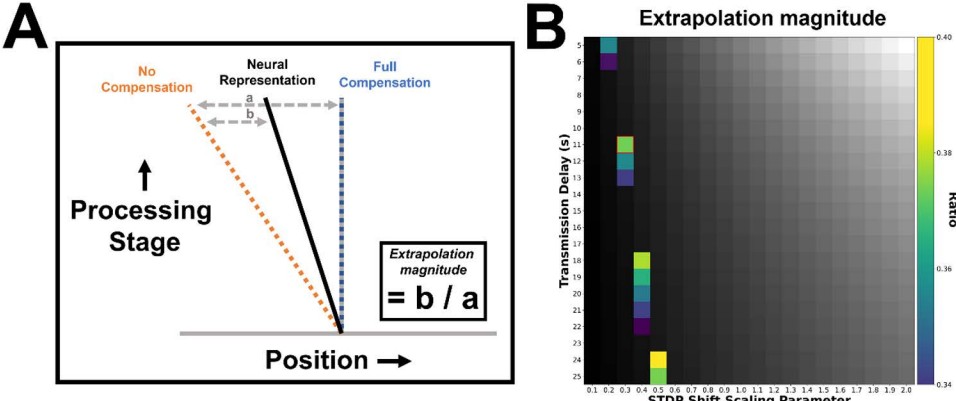

**Fig 6. Calculating extrapolation magnitude.** (A) Neural activity in the rotated position representation maps can be characterized with reference to the 'No Compensation' line, indicating where the stimulus would be represented in the absence of any extrapolatory mechanisms, and the 'Full Compensation' line, indicating the current stimulus position and therefore where the stimulus would be represented in the absence of delays. The extrapolation magnitude is quantified by the actual neural representation (determined by the peak position of stimulus-driven activity), as a ratio of the distance between the No-Compensation and Full-Compensation lines. A single measurement of extrapolation magnitude was made at the final time point of the stimulation, in order to allow enough time for activity to propagate fully throughout the network and for the peak of neural activity (relative to the stimulus) to stabilize. (B) Extrapolation magnitudes measured while varying the transmission delay and the STDP shift magnitude scaling parameter ($\beta$). Values within the range measured in the EEG analysis (.34 to .40) are color coded. Values outside this range are shown in grayscale. The point in this parameter space used in the main analyses (Fig 4) is highlighted in red.

the EEG data. Fig 6B shows the values of delay time and STDP shift scaling parameter which produced extrapolation magnitudes most closely matching the EEG data. In line with Alamia and VanRullen [34], we ultimately constrained the network to have an 11 ms transmission delay (such that with recurrent connectivity it would, in principle, generate oscillatory activity in the alpha/low-beta range) and took the best fitting value of $\beta$ within this row, for use in the subsequent analyses ($t_{delay}$ = 11 ms, $\beta$ = .3).

## Supporting information

**S1 Fig. Statistically thresholded maps.** From left to right, the panels show group-level spatio-temporal maps centered around motion onset, offset, and reversal. Diagonal black lines mark the true position of the stimulus. Horizontal dashed lines mark the time of the event of interest (stimulus onset, offset, or reversal). Red indicates high probability regions and blue indicates low probability regions relative to chance (0.025). The raw data are plotted with low opacity. Semi-transparent overlays denote positions where one-sample t-tests (one-tailed) against chance were significant at thresholds of $p < .05$ (medium opacity) and $p < .01$ (full opacity), respectively.
(TIFF)

**S2 Fig. Replication of main latency shift using the distribution centroid as a discrete position estimate.** To ensure that the main latency shift effect is robust across specific analysis choices, we re-plotted Fig 4A and 4B overlaying the centroid (i.e., vector average), instead of the point of peak probability, as a discrete position estimate. Examining S2 Fig, the same effect can be observed as we report in the main text, building confidence that this does not depend on the specific read-out method we choose to employ. In the main text we use the peak probability estimate as a more conservative read-out method, since this does not display the same extrapolative properties as the centroid during early processing (see Fig 3). All plotting conventions are the same as in Fig 4.
(TIFF)

**S3 Fig. Examining prediction formation in empirical and model position representations.** The top two rows show the temporal evolution of position information in the 250 ms following stimulus onset in the EEG data (top row: actual data, bottom row: synthetic control data). Bottom two panels show the equivalent for simulated network activity (top row: trained STDP model, bottom row: untrained control model). In the main analysis we consider the 'steady state' of position representations formed after sustained exposure to motion along a predictable trajectory, and found evidence of progressive extrapolation. However, after the initial appearance of a moving object, position representations must necessarily lag, since motion extrapolation is only possible once the object's velocity has been established. Here, we conduct an additional exploratory analysis to examine how rapidly motion extrapolation arises when a moving object first appears, and how the temporal evolution of this effect may be accounted for in the STDP network model. In S3 Fig we compare how both decoded and simulated positional representations evolve over timepoints immediately following stimulus onset. Initially, the decoded maps generated from the raw and synthetic EEG data (top panels) are similar. However, from ~150 ms the bulk of the high probability region in the raw map begins to shift forwards, with only a small portion of activity left traveling diagonally along the No-Compensation line. No such shift occurs in the synthetic map, with activity remaining centered on the No-Compensation line. This indicates that it takes ~150 ms for the 'steady state' temporal shift which we observed after sustained exposure to smooth motion to emerge. For the simulated maps, we can see that the same forwards shift occurs in the population-level activity of the trained STDP model, but not the untrained (control) model. In the trained model peak activity initially following the No-Compensation line, but then gradually shifts forwards across later timepoints. This occurs because of the velocity estimation process, in which the individual weights for each velocity sub-population change as a function of time. At the earliest timepoints, all subpopulations are assumed to be active. Taking the average of each of these yields a global representation in which the constituent STDP shifts are effectively canceled out (given that the range of velocities is symmetric around zero). As the information about the stimulus velocity is integrated,

the global response gradually becomes dominated by the subpopulation tuned to 1 cycle/s, and the represented position shifts according to the STDP shift magnitude associated with this velocity. This shift in the global position representation occurs at each layer in a time delayed fashion, causing the angle of the line tracking activity peaks to change across layers during the intermediate timepoints. At the later timepoints, all layers have received full information about the stimulus velocity, resulting in a straight line connecting the activity peaks. While this qualitatively captures the dynamics in the EEG-derived maps, it is important to emphasise the speculative nature of this additional analysis and assumed velocity estimation mechanism. These hinge on the assumption that all velocity-tuned subpopulations are activated by the onset of the stimulus (i.e., the onset transient), with the population tuned to the true velocity eventually winning out (either due to direct competition, or a passive fading of onset evoked activity). However, it is also possible that no velocity-tuned populations are initially activated by stimulus onset, and that activity simply gradually builds up in the population tuned to the true velocity. Arbitrating between these two possibilities remains an interesting avenue for future research that will likely require the use of direct intracranial recording techniques. Crucially, the assumed velocity estimation process only alters the early temporal dynamics of the network, and has no influence on its ultimate 'steady state' behavior. As such the main simulations are independent of this specific component/assumption, and only rely on the reasonable assumption that neurons tuned to a specific velocity are active after sustained exposure to motion.
(TIFF)

## Author contributions

**Conceptualization:** William Turner, Charlie Sexton, Philippa A. Johnson, Hinze Hogendoorn.

**Data curation:** William Turner, Charlie Sexton.

**Formal analysis:** William Turner.

**Funding acquisition:** William Turner, Hinze Hogendoorn.

**Investigation:** William Turner, Philippa A. Johnson, Ella M. Wilson.

**Methodology:** William Turner, Ella M. Wilson.

**Project administration:** William Turner, Ella M. Wilson.

**Software:** William Turner, Charlie Sexton, Philippa A. Johnson.

**Supervision:** Hinze Hogendoorn.

**Visualization:** William Turner, Charlie Sexton.

**Writing – original draft:** William Turner, Charlie Sexton.

**Writing – review & editing:** William Turner, Charlie Sexton, Philippa A. Johnson, Ella M. Wilson, Hinze Hogendoorn.

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
