## [Editor Report · Decision Letter 0]

9 Jan 2025

Dear Dr Turner,

Happy New Year!

Thank you for submitting your manuscript entitled "Progressive multi-stage extrapolation of predictable motion in human visual cortex" for consideration as a Research Article by PLOS Biology.

Your manuscript has now been evaluated by the PLOS Biology editorial staff as well as by an academic editor with relevant expertise and I am writing to let you know that we would like to send your submission back to external peer review, despite some reservations whether the reviewers' concerns have been fully addressed.

Once your full submission is complete, your paper will undergo a series of checks in preparation for peer review. After your manuscript has passed the checks it will be sent out for review. To provide the metadata for your submission, please Login to Editorial Manager (https://www.editorialmanager.com/pbiology) within two working days, i.e. by Jan 11 2025 11:59PM.

Kind regards,

Christian

Christian Schnell, PhD

Senior Editor

PLOS Biology

cschnell@plos.org

---

## [Decision Letter · Decision Letter 1]

7 Mar 2025

Dear Dr Turner,

Thank you for your patience while we considered your revised manuscript "Progressive multi-stage extrapolation of predictable motion in human visual cortex" for consideration as a Research Article at PLOS Biology. Please allow me first to apologize for the long delay in sending our decision. Unfortunately, one of the original reviewers was not able to submit their report after agreeing to re-review your manuscript, so we had to find a replacement reviewer. Your revised study has now been evaluated by the PLOS Biology editors, the Academic Editor, one the original reviewers and one new reviewer. Please note that Laurent was Reviewer 2 of the original submission, so Reviewer 2 of this submission is the replacement reviewer for Reviewer 1 of the original submission.

In light of the reviews, which you will find at the end of this email, we are pleased to offer you the opportunity to address the remaining points and comments from the reviewers in a revision that we anticipate should not take you very long. We will then assess your revised manuscript and your response to the reviewers' comments with our Academic Editor aiming to avoid further rounds of peer-review, although we might need to consult with the reviewers, depending on the nature of the revisions.

You will see that Reviewer 1 only has a few minor concerns and that Reviewer 2 is mostly satisfied with your response to Reviewer 1's original concerns, but there are a few items that need further addressing. We also encourage you to address the additional points that are raised by Reviewer 2. Please do not hesitate to get in touch if you have any questions or need a bit more time.

**IMPORTANT - SUBMITTING YOUR REVISION**

*Resubmission Checklist*

*Published Peer Review*

*PLOS Data Policy*

*Blot and Gel Data Policy*

Sincerely,

Christian

Christian Schnell, PhD

Senior Editor

PLOS Biology

cschnell@plos.org

REVIEWS:

Reviewer #1 (Laurent U Perrinet): I would like to thank the authors for the new version of their manuscript, as well as for the detailed response to our comments. These two elements allow me to judge that this article is now acceptable for publication in this journal.

I have just a few minor points that could be included in the final version.

- The choice of decoding score (point R2C12) can be simplified by giving it a physical meaning. By using a value between -1 and 1, we are in fact calculating the cosine of the difference between the predicted angle and the actual angle. It may be simpler simply to give this angle difference.

- In Fig 2A, you could mention that the "hockey club curve" is similar to that observed in the Frohlich effect (see for instance work from Jancke) - this would also allow you to justify the fact that the temporal resolution here is not enough to conclude for a "jump" at reversal in panel C

- in point R2C7 evidence is defined as the posteriori probability however position eviidence (eg in Fig2 or 3, 4) can be negative. please provide a clear definition in the methods

Reviewer #2: I stepped in for R1 (as a new reviewer) after a first round of revision.

In my reading, the main concern of R1 is that reported effects may have oculomotor origins rather than visual-processing properties inherent in the visual system. I do agree with the original R1 that it would have been more convincing if the authors had included careful eye-tracking when collecting their data (simply assuming participants follow instructions to fixate may be too optimistic).

In response to this main first concern of R1, the authors now make the case that their findings are not driven by muscle artifacts by the eyes because their decoding localizes to posterior electrodes. While I agree this rules out the contribution of muscle artefacts associated with eye movements, it does not necessarily rule out potential secondary consequences of eye movements (moving the visual stimuli over the retina) that may also contribute to the decoding from visual areas. Thus, just because the decoding is posterior, it may still reflect a secondary consequence of eye movements (or micro eye movements), rather than reflecting a computation inherent to the visual system. This could possibly be acknowledged (or provided further evidence against). That said, I find the additional arguments convincing, such as how their decoder was trained on a case with static stimuli, and tested on a case with predictably moving stimuli.

I also agree with R1 that the data presentation could benefit from further statistical substantiation. For example, while figure 3 is extensively described on pages 7-8, there appears no statistical quantification to back things up for this point (and the same appears to apply for figure 2). This remains a bit odd, also to me.

In addition to the above points, R1 made several other good (and critical) comments, but I do not see these as major breaking points, and the authors' responses appear reasonable to me.

As a new reviewer brought in at this stage, reading the work did inevitably prompt some new points that I would like to share. However, I wish to refrain from listing these as major concerns, also because I see as my primarily role to evaluate the revisions done in response to original R1, as I reflected on above. I nonetheless hope my additional comments may be perceived as useful.

I was wondering the following about the data in figure 3. Given models were trained in response to localizer stimuli with sudden onsets, is it not trivial that the decoded data differ when applying the model to the onset of the motion (similar to the onset of the localizer) versus mid-motion (that is more distinct from the localizer, given single localizer stimuli were never presented for this long)? This difference (in figure 3) could thus either reflect a change in how predictable motion is represented (as currently interpreted) but may also be driven by an increasing mismatch between localizer and test data (when comparing onset vs. mid-motion) in terms of time from stimulation onset? (It appears to me that to rule this out, the authors would have had to also include long-duration localizer stimuli, and train the decoding model to both early-onset and mid-way points in the localizer data). I also note how the reported difference appears to be largely driven by a change in magnitude (more so than in reconstructed position; given the peaks appears to remain largely unaffected).

Also, if the primary findings reflect those in figure 4, why do we only see the data for mid-motion in this figure? Would it not be of central interest to also show a temporal profile starting from motion onset to see the development not only with respect to the time after localizer onset (in training data), but also the time after motion onset (in testing data)?

Finally, at various key instances, such as in the title and the abstract, the authors speak of "processing stages". At these key moments, it is not always sufficiently clear whether the authors mean 'brain area in the visual hierarchy' (V1 vs. MT), 'time from motion onset' (onset vs. mid-way), or something else. Upon further reading, it appears these terms actually reflect time after localizer onset and how this time varying localiser location signal predicts the location of a moving stimulus. Translating this operationalization to processing stages and hierarchy appears less trivial than the initial intuition I had when these terms were introduced. I wonder whether this could/should be better positioned up front, to avoid readers potentially feeling misled by the promise set up at the outset of this paper.

---

## [Decision Letter · Decision Letter 2]

17 Apr 2025

Dear Dr Turner,

Thank you for your patience while we considered your revised manuscript "Progressive multi-stage extrapolation of predictable motion in human visual cortex" for publication as a Research Article at PLOS Biology. This revised version of your manuscript has been evaluated by the PLOS Biology editors, the Academic Editor and one of the original reviewers.

Based on the reviews and on our Academic Editor's assessment of your revision, we are likely to accept this manuscript for publication, provided you satisfactorily address the following data and other policy-related requests:

* We would like to suggest a different title to improve its accessibility for our broad audience:

Predictable motion is progressively extrapolated across temporally distinct processing stages in the human visual cortex

* Please include information in the Methods section whether the study has been conducted according to the principles expressed in the Declaration of Helsinki.

* DATA POLICY:

* CODE POLICY

We expect to receive your revised manuscript within two weeks.

*Published Peer Review History*

*Press*

Sincerely,

Christian

Christian Schnell, PhD

Senior Editor

cschnell@plos.org

PLOS Biology

Reviewer remarks:

Reviewer #2: The authors have done a great job in responding to my comments. I have no further comments.

---

## [Editor Report · Decision Letter 3]

30 Apr 2025

Dear Dr Turner,

Thank you for the submission of your revised Research Article "Predictable motion is progressively extrapolated across temporally distinct processing stages in the human visual cortex" for publication in PLOS Biology. On behalf of my colleagues and the Academic Editor, Christopher Pack, I am pleased to say that we can in principle accept your manuscript for publication, provided you address any remaining formatting and reporting issues. These will be detailed in an email you should receive within 2-3 business days from our colleagues in the journal operations team; no action is required from you until then. Please note that we will not be able to formally accept your manuscript and schedule it for publication until you have completed any requested changes.

PRESS

Sincerely, 

Christian

Christian Schnell, PhD

Senior Editor

PLOS Biology

cschnell@plos.org